# Communities’ Perception, Knowledge, and Practices Related to Human African Trypanosomiasis in the Democratic Republic of Congo

**DOI:** 10.3390/diseases10040069

**Published:** 2022-09-26

**Authors:** Charlie Kabanga, Olaf Valverde Mordt, Florent Mbo, Medard Mbondo, Donatien Olela, Rinelle Etinkum, Dieudonne Nkaji, Bienvenu Mukoso, Lubanza Mananasi

**Affiliations:** 1Independent Consultant, Welwyn AL6 0DS, Hertfordshire, UK; 2Drugs for Neglected Diseases *initiative*, 1202 Geneva, Switzerland; 3Drugs for Neglected Diseases *initiative*, Quartier Socimat, Gombe, Kinshasa, Democratic Republic of the Congo; 4Independent Consultant, Quartier Ngansele, Mont Ngafula, Kinshasa, Democratic Republic of the Congo; 5Faculty of Social, Politic and Administrative Sciences, Department of Sociology, University of Kinshasa, Kinshasa P.O. Box 127, Democratic Republic of the Congo; 6Deutsche Gesellschaft für Internationale Zusammenarbeit, Gombe, Kinshasa P.O. Box 7555, Democratic Republic of the Congo; 7Faculty of Social, Politic and Administrative Sciences, Department of Anthropology, University of Kinshasa, Kinshasa P.O. Box 127, Democratic Republic of the Congo

**Keywords:** community, HAT, knowledge, screening, treatment, control, participation, social

## Abstract

Background: The number of human African trypanosomiasis (HAT) cases in the Democratic Republic of Congo (DRC) has significantly reduced, thanks to more effective drugs and screening tools and regular mass screening. However, this potentially jeopardizes HAT control activities, especially community engagement. Methods: We used an ecological model framework to understand how various factors shape communities’ knowledge, perceptions, and behavior in this low endemicity context. Community members, frontline health providers, and policymakers were consulted using an ethnographic approach. Results: Communities in endemic areas are knowledgeable about causes, symptoms, and treatment of HAT, but this was more limited among young people. Few are aware of new HAT treatment or screening techniques. Participation in mass screening has declined due to many factors including fear and a lack of urgency, given the low numbers of cases. Delays in seeking medical care are due to confusion of HAT symptoms with those of other diseases and belief that HAT is caused by witchcraft. Conclusions: Community members see their role more in terms of vector control than participation in screening, referral, or accepting treatment. We propose recommendations for achieving sustainable community engagement, including development of an information and communication strategy and empowerment of communities to take greater ownership of HAT control activities.

## 1. Introduction

Human African trypanosomiasis (HAT) is a disease endemic in Sub-Saharan African countries, including the Democratic Republic of Congo (DRC), where the highest number of cases is found. This neglected tropical disease (NTD) can lead to irreversible deterioration of the nervous system and/or death if not treated early. In DRC, pre-independence eradication efforts almost led to HAT elimination in 1960 [1], but due to a weakened monitoring system, the number of cases increased again, reaching their highest level in 1998. Current collaboration between the Ministry of Health, donors, and national and international organizations has brought new cases to their lowest level yet (in 2020 only 395 new cases were reported in DRC, which remains the most affected country).

Control efforts included social research, which contributed to understanding communities’ behavior during high prevalence periods, but such research is lacking for the present low prevalence context. An understanding of local communities’ practice and perceptions in relation to HAT, and the social-cultural factors that influence their behavior, is needed to develop strategies that target the elimination of HAT in DRC, by enhancing local community engagement with screening, treatment, and prevention in this low prevalence context.

### 1.1. The Disease

HAT, or sleeping sickness, is only found in Sub-Saharan Africa, where it is endemic [2]. It is caused by two types of *Trypanosoma brucei* (*T.b.*) parasites: *T.b. gambiense* and *T.b. rhodesiense*, which are transmitted by the tsetse fly. After an infected bite, the parasite progresses through the body. First, the parasite is found in the blood and lymphatic system, causing headache, fever, weakness, pruritus, and muscular pain. These early symptoms are often confused with other diseases such as malaria. Later, the parasite crosses the blood–brain barrier and affects the central nervous system, resulting in the most specific symptoms of HAT: sleep disorders, abnormal movements, mental disturbance, and antisocial behavior [2].

*T.b. gambiense* is the parasite responsible for the chronic form of HAT found in DRC [2]. It can take several years for infected people to develop symptoms associated with this disease. While there has been a lot of effort to fight the disease in the 22 endemic provinces, the prevalence in 6 provinces (Kasai, Kasai Central, Lomami, Kasai Oriental, Kwilu, and Maindombe) (see Figure 1) is still a concern for the National Sleeping Sickness Program (PNLTHA) and other actors involved in fighting HAT [3]. 

HAT control includes three key strategies: screening of people living in endemic areas, treatment of infected people to eliminate the human reservoir, and the reduction of human–tsetse fly contact through vector control activities [4,5]. Screening for HAT is done actively or passively. Active screening includes mass campaigns organized by mobile teams aimed at all people living in targeted endemic areas. Passive screening happens at health centers or hospitals when infected people seek health care. Treatment of HAT is offered free of charge to those who have been infected [6,7]. 

### 1.2. Local Communities’ Knowledge of and Behaviors towards HAT

In DRC, local communities living in endemic areas have a good understanding of HAT, including the vector and the symptoms (especially for the second phase). They also have good information about control measures: diagnosis, treatment, and vector control [8]. In DRC, most community members in endemic areas are knowledgeable about screening processes, especially lumbar puncture and its side effects [8]. When it comes to treatment, people living in endemic areas know that sleeping sickness can only be treated by specialized health workers [8]. People from the community can tell the difference between Arsobal (the old treatment) and NECT (the newer treatment) [9]. In relation to vector control, local communities are familiar with the main strategies: tsetse traps and clearing their surroundings, which contribute to reducing tsetse fly–human contact [10]. This local community knowledge of HAT has also been found in studies conducted in other endemic countries such as South Sudan [11], Uganda, and Kenya [12]. However, in DRC, as in other African countries, beliefs linking HAT (especially the symptoms of the second phase) to witchcraft or dark forces still persist, in spite of years of health education [12,13,14,15]. 

The behavior of local communities in relation to HAT control activities have been documented over the years. In DRC, during the colonial period, local communities participated in screening activities in large numbers as it was compulsory, and those who did not participate ran the risk of being arrested [5]. During the recent HAT outbreak (1990s), participation in screening was also good, but there were still those who did not want to be screened for various reasons e.g., fear of stigma, fear of being diagnosed with HIV [14], fear of undergoing treatment (especially lumbar puncture), and reluctance to observe taboos [3]. Over time, participation in mass screening campaigns organized by mobile teams has decreased in endemic regions, due partly to the decrease in numbers of new cases [8]. Behavior towards treatment is also linked to screening, because HAT treatment used to be feared and people tried to avoid it for many reasons: the length of the treatment, fear of the side effects of Arsobal, the need to respect the various prohibitions or taboos during treatment (not eating hot food, having sexual intercourse, walking in the sun, or doing hard labor), having to undergo treatment away from the family, and the opportunity costs that come with loss of income [9]. Improvement of the drugs has made HAT easier to manage, but people living in endemic areas are still reluctant to accept treatment because many are not informed about the new simple regimen. As a result, many people living in endemic areas of DRC do not participate in screening in order to avoid treatment [3]. 

### 1.3. The Socio-Ecological Model for Analysing Factors Contributing to HAT Health Behaviors

The socio-ecological model is a useful one for health professionals and policymakers who are seeking to promote positive health behaviors. Short and Mollborn (2015) [16] define a health behavior as “actions taken by individuals that affect health or mortality. These actions may be intentional or unintentional and can promote or detract from the health of the actor or others”. The socio-ecological model is one of the frameworks that was developed in response to criticism of the “health belief model”. This conceptual framework was criticized for ignoring other factors contributing to health behaviors beyond individual choices and motivations [17]. The socio-ecological model enables understanding of the multifaceted nature of health behaviors [16] by considering individual-, interpersonal-, and societal-level factors. 

Several studies have been done on social factors that influence local communities’ behaviors toward HAT [3,9,18]. However, few have specifically used a socio-ecological model. For instance, Mpanya [19] used the “Precede-Proceed Model” to study socio-cultural factors that influence local populations’ attitudes and practices toward HAT in DRC. The different levels (individual, interpersonal, and societal) of factors are found in different studies, although separately. At the individual level, the health situation can influence whether a person takes part in HAT screening or not, with those feeling ill more likely to participate [20].

At community or interpersonal levels, bringing screening services close to communities through mass screening increases the chance of people being screened, but factors such as the mistrust of health service providers, especially the mobile team who are believed to inject diseases, deters community members from participating in screening [21]. Being far from their family during treatment (as most treatment centers are found in larger hospitals) is another interpersonal factor that negatively affects attitudes toward treatment, because the social network of family and friends plays a major role in caring for the sick [11]. 

HAT patients are perceived as being unable to fulfil certain social expectations, such as caring and providing for their families, being able to reproduce (as HAT affects fertility), or being rational, so they are stigmatized or discriminated against [3]. As a result, many people avoid getting screened or treated. HAT is also mostly prevalent in rural areas where the people are unable to afford the medical costs related to HAT treatment. By providing HAT treatment free of charge, PNLTHA has been able to remove one of the major obstacles to treatment [6]. However, the indirect costs (transport, food, and living costs for the carer), as well the opportunity costs (when the patient is unable to work while undergoing treatment), have remained major impediments to the acceptance of treatment [11]. Endemicity also seems to affect local communities’ behavior. With the reduction in numbers of HAT, the perceived risk posed by the disease decreases, and as a result, local participation in screening and vector control activities reduces [22] and local leaders and community volunteers demand incentives to organize HAT control activities [5]. At a policy level, many social factors that affect behavior, such as stigma, have not been sufficiently addressed in the implementation of HAT control activities. During mass screening, people are not provided enough privacy [9]. Although HAT affects patients and their families psychologically, HAT is one of the NTDs where the management approach focuses on medical treatment but overlooks personal support and counselling of patients, community education about stigma, or other negative experiences resulting from this disease [23].

The aim of this study was to ascertain the current level of knowledge and the attitudes of local communities (patients, local and traditional healers, local health service providers, and other community members) from six endemic provinces of DRC (Kwilu, Maindombe, Kasai, Kasai Oriental, Kasai Central, and Lomami) in relation to HAT. In addition, a socio-ecological model was chosen to identify the factors influencing positive health behavior from a multilevel perspective. The study hypothesis was that local communities’ HAT-related knowledge and practices were influenced by socio-cultural representations of the disease.

## 2. Materials and Methods

The study was conducted using a qualitative method, based on an ethnographic approach. The main data collection activities were focus group discussions (FGD), key informant interviews (KII), participatory workshops (PW), life story or in-depth interviews (IDI), and observation. Research participants were selected at three levels: local or community, provincial, and national (Table 1). They came from all intervention health zones selected for the full project among those with higher HAT endemicity (see Figure 2).

FGDs were organized with community members: farmers, fishers, teachers, traders, etc., and children and teenagers. To create an environment in which participants felt free to express themselves, in most cases, there were separate focus groups for men, women, female adolescents, and male adolescents. In some cases, however, mixed groups were organized. FGD guides were designed to gather information related to the community’s representation of HAT (the tsetse fly and HAT health service provision) as well their knowledge and experience of HAT management. Similarly, PWs were also organized for community members. PWs are a type of focus group discussion but are more structured. Participants were asked to give their views (agree, disagree, or neutral) about a statement that reflected an attitude or knowledge or behavior related to sleeping sickness. Participants, who were grouped based on their views, were encouraged to debate and explain why they held that view. The PWs allowed for the gathering of rich information from the interaction between participants. Additionally, researchers were able to get a rough estimate of the proportion of people holding a specific view. Using other activities, such as drawing during PWs, gave participants a creative way to express themselves, especially when it came to giving recommendations on what they felt should be done to address the issues identified. The PW guide was designed mainly to gauge local communities’ perceptions and beliefs in relation to HAT. FGDs and PWs lasted on average between 40 and 60 min. For each data collection activity, care was taken to ensure that gender issues were captured.

IDIs or life stories were organized with people who had been directly affected by HAT, i.e., current or previous patients, and relatives of current or deceased HAT patients. The life story guide was designed to gather participants’ perspectives and experiences of the disease. It also aimed at understanding the communities’ perspectives on the reasons for the death of HAT patients who did not recover. Life stories lasted between 25 and 40 min.

KIIs were done with community leaders (church leaders, traditional chiefs, political officials, etc.), local health services providers (traditional and formal), and provincial and national focal points for HAT, as well as focal points of NGOs working on NTDs. The focus of the KII was the perception of HAT (past and present), perspectives on community knowledge and practices related to HAT, barriers to and reasons for local community engagement, and the status of the HAT EIC. In addition, the guide for provincial and national participants focused on policy-related issues, especially in relation to the elimination of the disease in DRC. KIIs lasted on average 30 min. 

Observation was embedded in all data collection activities, but a guide was designed to gather information related to exposure and risk of contamination by HAT. Key issues that were observed included locations with high risk of tsetse fly bites (water collection points and markets), general community practices related to vector control, and health seeking practices in non-formal or private health provision (churches, traditional healers, diviners, pharmacy, market, etc.).

All the guides were translated into local languages (Kikongo, Tshiluba, and Lingala), and data collection activities were done in these languages, except in some cases when other local dialects were used for optimal comprehension. KIIs were conducted mostly in French. A multidisciplinary team of seven people (social researcher, anthropologist, two medical doctors, and a sociologist) was responsible for data collection. Each member was assigned two health zones based on their knowledge of the local language and their experience of the region. The principal investigator was also involved in data collection. Assistants were recruited locally and were responsible for organizing meetings and/or translation when needed. 

To ensure that the data collection tools addressed the needs of the study, the guides were piloted in areas neighboring Kinshasa (Bibua and Maluku). The team was deployed in two phases. In the first phase, a team of two people was deployed to two communities and the guides were adjusted based on their observations. In the second phase, the remaining 12 communities were covered. 

The research team started with community-level data collection, which informed adjustments to the questions that were prepared for provincial- and national-level participants. Data were collected using a digital tool. At the end of data collection, the audio files were transcribed and translated in a Word file. Translations were verified by sampling transcripts and asking the data collectors to confirm whether the translation was in line with what they had done in the field. 

Participants were accessed through local health centers and with the assistance of community health mobilisers. Inclusion criteria were having lived in the community for the last two years (community members) and having worked on HAT in the area for at least 6 months (key informants). 

### 2.1. Data Analysis 

After the interviews were transcribed and translated, the research team performed qualitative analysis of the data. After the data were classified, the interview content was coded according to themes relevant to the research questions and conceptual models (cognitive and socio-ecological), using NVivo qualitative analysis software.

The qualitative data analysis was guided by two models, namely the health belief model [24] and the socio-ecological model [25]. In addition to these two models, the research team also used the gender approach and the community engagement guide [26].

### 2.2. Ethical Consideration 

The study was approved by the Ethics Committee of the School of Public Health of the University of Kinshasa, DRC. Informed consent was also obtained from all subjects involved in the study. Community members gave oral consent and agreed to sign the attendance list (with a pen or thumb print), while key informants provided written consent.

## 3. Results

A story about the experience of HAT:


*I am the wife of the late Shabana (the real name has been changed). We lived in Mbuji Mayi. My husband had made a trip to the village. It is there that he was bitten by the tsetse fly. When he returned, we noticed that he had become talkative, contrary to his nature. We thought it was malaria. We went to the hospital and they examined him and told us that it was not malaria. When we returned, his parents came to take him for treatment. We had to travel a long distance to the sleeping sickness treatment center. There, after having treated him, we observed that the chattering had stopped. He ate his food cold. He was not supposed to walk under the sun, so he used an umbrella or covered his head with a cloth if he needed to go somewhere. All day long we poured water on his head to allow the medicine to penetrate. After the treatment, we went back to his parents’ village. He had gained weight. His family advised me to go back home to Mbuji Mayi to rest. After that, he relapsed, and his family took him back for treatment. A month later, he relapsed again, and we had to return again for another treatment. But when we got there, my husband refused the treatment. We went everywhere without success. At the end he could no longer stand, we had to feed him. A year later he started having hiccups. We decided then to resort to the traditional treatment. Thank God the hiccups ended. He had regained some strength, but he was no longer able to talk. The suffering continued for a year and five months until he died.*


### 3.1. Communities’ Knowledge about and Perceptions of HAT

These types of stories about HAT may have become rare in many endemic areas, but they illustrate the experience of those who have been affected by the disease. They also give a snapshot of local communities’ knowledge, perception, and practices about HAT, which are presented in the following results section.

#### 3.1.1. The Disease

Historical perspective: HAT is not a new disease for people living in endemic areas in DRC. When community members describe the history of the disease in their community, they talk about it having affected their ancestors, being around during colonial times. This knowledge may partly be due to their exposure to information from health professionals. 

*‘The disease has been around since our ancestors’ time. According to our history, our ancestors were affected by sleeping sickness because they used to sleep all the time’* **Focus group with men in Kwilu**


*Question: Since when did you know that sleeping sickness exist?*


*Answer: We know that this disease exists since our ancestor’s time and from nurses who work for FOMETRO»* **Focus group with men in Kasaï Central**

FOMETRO (Fons Médical Tropical) was an NGO with Belgian government funding in charge of the support of Trypanosomiasis Central Bureau in DRC up to 2003 when the PNLTHA was restructured. Community members demonstrate a good knowledge of the current status of HAT. They seem to know that there is an improvement, as there are no or very few cases found in their communities. However, there are also many people who cannot say whether it no longer exists in their community, as information about cases that have been found is not shared with community members. 

The cause of the disease and vulnerability: Local communities know that the tsetse fly is the vector that transmits HAT. They understand the transmission cycle, showing how the fly that has bitten an infected person can transmit the disease to a healthy person. However, many beliefs still exist about other causes such as witchcraft, bad spells, and other forces:


*‘The sleeping sickness I suffered from was caused by a bad spell from my husband’s other wife who wanted him to divorce me.’*
**Former HAT patient, Lomami**


Community members perceive that everyone is at risk of catching the disease, regardless of their age group or gender.

*‘It is a disease that spares no one, men or women, children or old people, everyone can be affected by sleeping sickness.’* **Local leader, Central Kasai**

However, there is also a perception that some activities make groups more vulnerable to the disease. These vulnerable groups and the high-risk activities are different depending on the regions. 


*Question: Why do you think men are more likely to be stung?*


*Answer: It is because they work in the forest. Especially in the morning the tsetse fly bites the men in the forest and at the river where we work.* **Focus group with men, Kasai Central**

*‘We see that here it is the women who suffer more from sleeping sickness than the men. Because women go to the fields every time, this fact makes them more vulnerable; they go fishing where there is a strong presence of tsetse flies.’* **Focus group with women, Kwilu**

The symptoms: Local communities seem to have limited knowledge of the early symptoms (or stage 1) compared with later symptoms (or stage 2) of HAT. When asked how to recognize whether a person is suffering from HAT, community members mostly mentioned abnormal sleeping patterns (sleeping in the day and awake during the night), mental disturbance, unsocial behaviors, and loss or gain of weight.

*‘Someone can have earlier symptoms of sleeping sickness without knowing about it. He/she can be taken to the hospital or to the health center where they can start treating malaria, while the disease is getting worse until when the person starts to show signs of mental disturbance.’* **Focus group with women, Lomami**

Earlier symptoms of HAT, such as fever or headaches, are usually associated with malaria or typhoid, whereas second stage symptoms are said to be caused by witchcraft. 


*Question: When someone has fever and headaches can it be caused by trypanosomiasis?*


*Answer: No, it can be malaria. But if in addition to those symptoms, the person is having abnormal sleeping patterns, then we can say that it is trypanosomiasis.* **Participatory workshop with women, Mai-Ndombe**

*‘There was a boy who came from B…, he was also showing the symptoms you are talking about [change in sleeping patterns and unsocial behaviors], we thought this was not normal. He was taken to the hospital, and they found he was suffering from sleeping sickness. But for other people those symptoms are not necessarily caused by sleeping sickness, for instance women who want to control their husband through witchcraft, or men who do witchcraft to find a job, they can also get mental disturbance.’* **Focus group with women, Kasaï**

Screening: Knowledge about the screening processes for HAT is well established among communities in endemic areas. They know about mobile teams that have been performing active mass screening. 

*‘Since I have been living here sleeping sickness has always been present. I know this because FOMETRO come here every 3 months to screen.’* **Church leader, key informant, Kwilu**

When asked about the procedures for screening, community members describe lumbar puncture. They also know that mobile teams try to make the screening process confidential. Passive screening was also mentioned by community members, who know that health providers at the health center or main hospital can also identify a person suffering from HAT.

*‘They [mobile team] extract a liquid from your back with a syringe. This process can destroy families, especially for men as they can become impotent.’* **Focus group with Men, Lomami**

*‘There are people who refused to be screened because the procedure is painful, especially the injection in the spinal cord’* **Focus group with adolescent girls, Kwilu**

Most community members do not seem to be aware of the rapid diagnostic test (RDT).

Treatment: community members talk with ease about different steps after the screening, including the referral process and treatment.

*‘Everything starts with going to the local health center because at that point you don’t know what you are suffering from. But when the nurse does a test and finds out that you are actually suffering from sleeping sickness, he will transfer you to the Ngandajika [Main hospital]’* **Focus group with women, Lomami**

It is accepted by most informants that HAT is only treatable with modern medicine, and that no alternative medicine can cure HAT:

*‘It is impossible to cure this disease with traditional medicine, so to recover from it one needs to be treated with modern medicine’* **Focus group with women, Kasai**

Beliefs that recovering from HAT depends also on its source are still common; when its cause is “natural”, it can be treated at the hospital, but if it comes from supernatural forces, the patient cannot recover unless they use alternative treatment. While many people still talk about the treatment done through injection, referring to Arsobal, there are a few who know about the oral treatment, especially those who have been treated recently. 

*‘Some people don’t want to go to the health centers to get treatment. They usually say they may get an injection which will cause disability…’* **Focus group with men, Kwilu**

It is perceived that recovery from HAT depends on early diagnosis and on whether the cause of the disease is “natural” or a result of witchcraft.

*‘People die because they were diagnosed late. They will also die if their disease was caused by witchcraft. In that case, no matter what medical staff does, they will die’* **Focus group with adolescent girls, Kwilu**

Abiding to certain prohibitions is also perceived as essential for a person to be able to recover from HAT, even with the new treatment.

*‘The prohibitions must be followed even now; they will never end. The prohibitions do not change with the evolution of the treatment’* **Focus group with men, Kasai Oriental**

However, there are a few people who seem to think that it is no longer necessary to follow prohibitions, except the one related to sexual intercourse.


*‘But now there is no longer an injection, you can be treated with a pill, there are no longer many prohibitions…’*
**Focus group with men, Kwilu**


Difference in level of knowledge: The level of knowledge about HAT is heterogenous among community members. In general, the level of knowledge seems to depend on the experience of the disease, with previous HAT patients, their relatives, or friends having more detailed knowledge of the symptoms and treatment than the rest of the community. Former HAT patients or their relatives are one of the sources of information for those wanting to know more about HAT symptoms or treatment.

*‘There used to be a woman who was living not far from here, she had sleeping sickness and was scared to talk about it. But one day she came to see me with her relatives to ask me about the symptoms of this disease. I told them about headaches, fever, …. Then I told them that they should not be afraid because those who treated me will do the same for her.’* **Life story with a former HAT patient, Kasaï Central**


*Question: Since you were born, have you ever heard about the sleeping sickness?*



*Answer: No, we have never heard about this disease. We only know about tetanus.*



*Question: There are sometimes people who come in your village with big vehicles, they put tables and boxes outside. What do they talk about?*


*Answer: We thought that they were examining people looking for tetanus, they were working under the mango trees* **Focus group with children 9–10 years old, Kasaï**

It also seems that the level of knowledge depends on age, with adults and older members of the community having more knowledge than youth and children. Many young people mentioned that they had never encountered a person suffering from the disease, and other children still do not know why mobile teams visit their community. There did not seem to be any difference in the level of knowledge of HAT between women and men. 

#### 3.1.2. Participation in HAT Control Activities 

Screening: In the six endemic provinces, active screening by mobile units continues to take place. In general, it seems that participation of community members in screening activities has decreased.

*‘There is a decrease in the number of people who participate in [screening]. There is negligence. I usually condemn this negligence because there can be unknown cases.’* **Key informant, community leader, Kasai**

Community members are informed about the screening activities prior to the arrival of mobile units, but some people refuse to get screened or are unable to participate. This situation sometimes leads local leaders to force community members into participating. 

*‘There are people who refuse to be tested saying that the treatment is very painful, especially the injection in the back.’* **Focus group with adolescent girls, Kwilu**

Health-seeking behavior: The health-seeking behavior of HAT patients seems to vary between people, especially as the early symptoms are usually confused with malaria or go unnoticed. HAT patients who get diagnosed through active screening seem to have a shorter care path, as they usually get an early diagnosis. As soon as they are diagnosed with HAT, they are referred to the treatment centers, and so are often able to recover very fast. 

*‘I suffered from this disease [sleeping sickness] in my childhood, it was when white people were around, I was treated, and I recovered. But when the mobile team came again in 2003 or 2004, I was screened, and the diagnosis was positive. I did undergo treatment again.’* **Former HAT patient, Kasaï**

However, diagnosis is usually late for many patients who are diagnosed through passive screening. Most of the time they are already showing second stage symptoms of HAT. According to community members, the delay in health-seeking behaviors is due to the lack of knowledge about early symptoms and about many second stage symptoms. However, there are also those who consult local health centers but are not diagnosed with HAT. Beliefs in witchcraft delay health seeking.

*‘When we see someone who is saying things that don’t make sense, or who has behavior problems, it is difficult to immediately think that the situation is caused by sleeping sickness, especially in a context where there are no cases of this disease like here in our community. In such situations, we would think first about witchcraft or demons, which will make people seek a solution within the extended family, in churches or from a traditional healer. It is only when these options are exhausted that we can decide to take the patient to the health center.’* **Focus group with men, Kasaï Oriental**

Vector control activities: Vector control activities seem to be neglected. Local leaders as well as community members spoke about obstacles that would prevent the effective implementation of vector control activities. Regarding tsetse fly trapping, the obstacles are lack of traps in the Kasai region, destruction of traps set in forests or along rivers in the Bandundu provinces, and the non-participation of community members in monitoring the traps.

*“We place mini screens along the river to catch tsetse flies, but people say that we placed them to capture evil spirits that will decrease the number of fish in the river.”* **Key informant, HAT stakeholder, Kwilu**

#### 3.1.3. Communities’ Role in HAT Control 

Most community members living in endemic areas see their role in HAT control as mainly related to vector control. They do not perceive that participating in screening, accepting treatment, or referral of sick people is their responsibility.


*Question: Could there be a role for the community members in HAT control activities?*


*Answer: Yes. As you have just said, there is a lot of vegetation in the neighborhood, it looks like a forest. Community members need to weed and keep the neighborhood clean. This is the work that the community should do.* **Key Informant, Church Leader, Kwilu**

There seems to be a power imbalance in the partnership between local community members and health professionals, with health professionals perceived as a group of people who hold all the knowledge and who are supposed to give instruction. Local communities see themselves as passive recipients of HAT control activities. Local service providers also have the same perception of local communities.


*Question: Do you have any recommendations for health professionals regarding HAT treatment?*


*Answer: No there are no recommendations. It is them [health professionals] who are well placed to know what they should be doing. What recommendation can we give them?* **Life story, former patient, Lomami**

*‘They [community] don’t have a role to play [in the sensitization activities], on the contrary, they need to be sensitized.’* **Key informant, health provider, Kasaï**

There does not seem to be ownership of HAT control activities by community members in either the Kasai or the Bandundu regions. The type of contribution made by local communities that used to be taken for granted, such as community mobilization by local leaders, is no longer available for free. Local leaders and community volunteers are now expecting incentives to do sensitization activities. 

*‘There was a local dignitary who complained, saying “I have been asking community members to do sensitizations”, while those who ask them to do that job are paid, and have a lot of money. They are asking us to work for free. They should also provide us with an incentive.’* **Key informant, local leader, Kasaï**

### 3.2. Using an Ecological Model to Understand Enabling and Limiting Factors for Positive Health Behavior 

Knowledge, attitudes, and practice about HAT are influenced by biological, social, and environmental factors. An ecological model was used for these factors that contribute to positive or negative health behaviors in relation to HAT. These factors were organized at three levels: the individual level, the interpersonal and/or community level, and the societal level, see Figure 3.

#### 3.2.1. Individual Level 

The perception of the threat posed by HAT, health status, personal beliefs, experience related to HAT, sex, gender, and age seem to play a role in whether a person will participate in HAT control activities. 

Perception of the threat posed by HAT: For most community members, there has been a significant decrease in the number of HAT cases. As a result, some of them believe that the disease has been completely eradicated:

*‘There used to be a time when many people died from sleeping sickness in our community. Our traditional chief organized a ritual, and a goat was sacrificed to get rid of this disease. Since that ritual, the disease has disappeared from our community. I can confirm that this disease no longer exists here.’* **Participatory workshop with women, Lomami**


*Question: Does sleeping sickness exist in this village?*


*Answer: It no longer exists; I have never seen a prescription for sleeping sickness medicine.* **Local pharmacist, Kasaï**

There are also some people who believe the disease still exists, and the reason for this is that HAT cases may go unidentified as not everyone gets screened. 

*‘Sleeping sickness still exists in our community. If they do a proper screening, we are sure that they will find some cases. As you can see, we have many palm trees here [where tsetse flies live].’* **Focus group with women, Lomami**

Other people are not sure whether the disease still exists or not. This lack of certainty is mainly because information about new cases is never disclosed due to confidentiality. 

*‘Regarding cases, it is difficult to know because they [mobile teams] have never shown us a list showing the existing cases, and how many are left after their work. This is the only way of knowing whether there is a decrease in the incidence of this disease.’* **Focus group with men, Kasaï**

Personal beliefs (in witchcraft or modern medicine) and trust in health service providers: People hold various beliefs about the cause of the disease, which influences where they seek health care. Some people do not trust the mobile teams that are responsible for mass screening, and suspect them of working for evil forces:

*‘Most of the people run away, those who attend the screening are very few. Even when they visit schools, people run away, they say that the blood that is taken is used for magical purposes. This is why they don’t like to get screened.’* **Life story, adolescent former patient, Kwilu**


*Question: Why do you think that individuals affected by the disease don’t want to go to the hospital?*


*Answer: It is because of the inaccurate information that the population receives. There are rumors about mobile teams, that say that some people who go to the hospital end up dying.* **Focus group with Men, Kwilu**

Health status: Many people shared that HAT symptoms prompted them or their families to go to the mass screening organized by mobile teams, or to the hospital or health center to get checked. When these people are diagnosed with HAT, they usually agree to undergo treatment. 


*‘I was not feeling well. Especially as I didn’t know which stage my disease had reached. So, I had to gather all my courage to go to the screening’*
**Former HAT patient, Kasaï**


Personal experience and knowledge of HAT: those who have suffered from HAT before, as well as those who are close to them (relatives or friends), seem to be very likely to participate in screening activities: 

*‘As I know already about sleeping sickness, I cannot run away [when the mobile team come for screening].’* **Adolescent former patient, Kwilu**

Age: many health providers who participated in the study reported that young people are less likely to agree to participate in screening activities compared to adults:

*‘My main concern about the elimination of the disease by 2030, is mainly the loss of collective memory. As time passes, children will grow up who have not experienced this disease. So, there will be a time when negligence can happen, as they will be wondering whether this disease is real, but for their parents who lived when this disease was visible, they remember, that these are disciplined people, and more likely to participate in screening activities.’* **Key informant, Ministry of Health**

This concern from health professionals and policymakers was reinforced by how young people perceive the risk of being infected by HAT: 

*‘We have never heard of young people who have been diagnosed with sleeping sickness, it is mostly adult men and women who are most affected.’* **Focus group with adolescent girls, Lomami**

Sex and gender: According to many community members, gender and sex could be also a factor that can lead to people participating in HAT control activities. Women are perceived to outnumber men during mass screening activities. Women are also more likely than men to go to the health center or hospital if they are not feeling well.

*‘Men pretend when they are sick, they don’t take it seriously, they continue to do their business as if nothing was wrong. But women are more susceptible to disease, especially because of pregnancies, they visit health centers more often, whereas men can spend up to 10 years without knowing what a health center looks like.’* **Focus group with men, Kasaï Oriental**

*‘Men don’t like to go to the health center. It is already late when they decide to go. By that time, symptoms will be advanced, then it becomes difficult to treat them.’* **Focus group with women, Lomami**

However, community members also believe that the high status of men in their communities and their control over the family’s resources (as household head) contributes to men getting more care and attention when they are seriously ill compared to women. On the contrary, because women (especially wives) have less decision-making power, they can experience delays in seeking health care because they must wait for their husband’s decision.

*‘In a couple situation, when the husband is sick, his whole family will get mobilized, from the youngest to the oldest, they will demand to know the cause of the disease. The wife will be harassed and accused of witchcraft. But when the wife is sick, she can get the support from her husband only, the extended family will not care. We can say with confidence that, within the extended family, the wife’s diseases don’t weigh the same as the husband’s.’* **Focus group with men, Kasaï Oriental**

In addition, women seem to be likely to get blamed for their own sickness or their husband’s.


*‘For some families, they would think that the wife is guilty, or she is a witch, that is why she will get neglected.’*
**Life story, the husband of a former HAT patient, Lomami**


In some cases, HAT patients get abandoned by their partners. According to community members, women have a higher risk of being abandoned if they suffer from HAT compared to men: 


*Question: If it were you suffering from the sleeping sickness, do you think that you would have the same experience as your husband?*


*Answer: The way I know men, he would have said, I have done everything, but you are not recovering. Your family should take you back and take care of you.* **Life story with the spouse of a former HAT patient, Lomami**

*‘We love our wives, and when they get sick, we take them to the hospital, if it doesn’t work, we take them to the church and that’s the last step. But if it still doesn’t work, we send them back to their homes, because we can’t continue living with a crazy woman, especially if this illness is a result of an issue within her family.’* **Focus Group with men, Kasai**

#### 3.2.2. Interpersonal/Community Level

At the interpersonal or community level, factors that influence local community behaviors toward HAT include the level of health service provision (quality, cost, and distance), family, stigma, local leadership, education, and sensitization activities.

Quality of health service: People who live in endemic areas think that the quality of health service-related HAT management (screening and treatment) is good. They perceive that health professionals involved in HAT management are well trained, and that health facilities are well equipped to take care of patients: 

*‘Sleeping sickness is treated especially in Ngandajika where there is a hospital and specialized doctors. Because even my parents were always treated in Ngandajika.’* **Focus group with women, Lomami**

Their confidence in the ability of health professionals to provide effective service is shown when they speak about people who were able to recover from the disease once they agreed to undergo treatment:

*‘The treatment is good because there are positive results after I have been treated. When I was sick, I was screened by a team of specialists that came from Kinshasa. As the diagnosis was positive, I underwent treatment, and I recovered.’* **Former HAT patient, Kasaï Central**

However, mobile teams are not always trusted by local communities, as they are believed to inject HAT or other diseases when performing screening:

*‘When we hear about the mobile teams it makes us wonder whether these people who come to screen us don’t bring other diseases that they inoculate us with?’* **Focus group with men, Kasaï Oriental**

Local health professionals are often not trusted by communities to manage HAT, as they are perceived as not having enough skills or equipment. For such community members, HAT can only be treated in larger hospitals:


*Question: Why do you say that health professionals from B… are not able to treat the disease?*



*Answer: Because they don’t have equipment and medicine.*


*Answer: We have never seen them treating sleeping sickness.* **Participatory workshop with adolescent girls, Kasaï Oriental**

The quality of HAT health services is usually appreciated, but the timing of screening activities is often not appreciated. Many community members think that some people do not participate in screening because it happens in the morning when they are busy with other essential activities: 

*‘Not all people attend screening activities. They give priority to their work in the fields, where they go early morning, and they return only late whereas we must work in the morning’* **Key informant, HAT health provider, Kasai Central**

The cost of health services: Most local community members know that treatment of HAT is free, so they do not see that as a constraining factor for accessing care. Even though the treatment is free, the indirect costs incurred when undergoing HAT treatment are considered by many people as a constraining factor for both screening and treatment. These indirect costs include opportunity costs, as they must forgo work, especially when the family provider is sick, plus the costs related to transport and food for the patient and the carer:


*‘The most difficult experience while I was suffering from sleeping sickness was the fact that I had to have a period of rest while undergoing treatment. I had children for which I had to pay school fees, and some of them were in the last year of secondary school. Because I did not have any income, it was difficult to handle this situation. At that time, I was thinking more about my children than of the disease, because I was already feeling well. But because of all the prohibitions I had to adhere to for six months like not listening to the radio, no hard labor [I could not do anything]’*
**Former HAT patient, Man, Kwilu**


Access to health facilities: Access to health facilities is one of the main challenges for local community members in agreeing to HAT treatment. They talked about the long distance they must travel to get to HAT treatment centers, poor road conditions, the lack of means of transport, and the money to pay for it. In some areas, such as Kasai provinces, the patient may have to travel up to 100 km by train (which is usually slow, infrequent, and unreliable):

*‘The distance to health facilities is a serious problem for us. There are many challenges when a patient is referred to Lukalaba or Mbujimayi, especially for the carer.’* **Focus group with men, Kasaï Oriental**

Mass screening by mobile teams has, however, brought the screening service close to local communities, making it easy for them to be screened on a regular basis.

Family: The family can influence health behavior, as it constitutes a person’s safety net in a context like DRC where formal social security systems do not exist. In the case of HAT, family members were reported to mobilize the financial resources needed to get their sick people to health facilities where HAT was diagnosed, and they received treatment. 


*Question: When you were asked to take your husband to get treated to Mbujimayi, who took him there?*



*Answer: He was taken to Mbujimayi by his family.*
**Life story with the spouse of a former HAT patient, Lomami**


However, family members are also perceived as causes of delays in seeking health care, especially those that believe that illness could be a result of witchcraft or demons. These families are reported as spending time trying to resolve family feuds, or taking members suffering from HAT to traditional healers or churches instead of to health centers:


*‘We have noticed that there are families that take their sick members to diviners, traditional healers, instead of taking them to the hospital.’*
**Key informant, female community leader, Kwilu**


Community attitudes toward HAT patients: Being associated with HAT results in stigma, as many people perceive HAT as a disease that affects the physical, mental, and social capacity of those who suffer from it. Those who suffer from HAT risk losing their social status. A good number of people who participated in the study reported that mockery and gossiping were one of the reasons people did not want to be screened: 

*‘Those who are suffering from sleeping sickness cannot share about it because other people would make fun of them. They can only talk about it with those who are close to them. You know it is a dangerous disease that can cause death. There is a risk of remaining crazy. It is a disease that you can compare to AIDS.’* **Focus group with men, Kasai Oriental**

Mobile teams try to keep the information about those who had a positive diagnosis confidential, but most people feel that there is limited privacy in the screening process as these activities usually take place in open spaces. They talk about people being taken to the “side” after the screening as those who might have the disease. Many former patients were aware of the discrimination they continued to experience even after recovering from the disease. 

*‘When playing with my friends, if I do anything that seems strange, they say that I have never recovered from the disease I suffered from. And if I do something that they don’t like, they say that my behavior is an after-effect of the sleeping disease’* **Former HAT patient, adolescent, Kwilu**

Local leadership: For many community members, the commitment of local leaders is key to people participating in HAT control activities. For instance, many people seem to think that the limited involvement of local communities in vector control activities is due to the lack of commitment of traditional chiefs or local officials in these activities: 


*Question: Why do you think the community doesn’t get involved in HAT control activities?*


*Answer: It is a leadership issue. Local chiefs may ask people to clear the area to keep it clean, but they don’t do any follow up to ascertain whether the work was done or not. There is no sanction for those who don’t follow the instructions.* **Focus group with men, Kasaï Oriental**

Health professionals decry the decrease in the involvement of local leaders, who are demanding to be compensated for their collaboration. 

Education and sensitization activities: Many community members living in HAT endemic areas reported that they were able to attend mass screening activities after hearing messages through megaphones: 


*Question: Before suffering from this disease, had you heard about it from somewhere?*


*Answer: No, except when the teams from FOMETRO came. We were asked to go to get screened.* **Former patient, Kwilu**

#### 3.2.3. Societal Level 

At the societal level, factors that were found to influence local communities’ attitudes and behaviors toward HAT control activities included the social representation of HAT, cultural and social norms, the context of HAT endemicity, the economic situation and source of livelihood, and policies and strategies related to community engagement.

Social representation of HAT: Local communities share certain perceptions about sleeping sickness. It is seen as a dangerous, deadly, and debilitating disease:

*‘When you mention sleeping sickness, the first idea that comes to our mind is suffering and death’* **Focus group with women, Lomami**

*‘Sleeping sickness is a very dangerous disease, if you are diagnosed with it, you need to go to the hospital where you can receive treatment, and it is free. You need to protect yourself against this disease.’* **Life story with a HAT patient, Kwilu**

The disease brings negative emotions, such as suffering, shame, and fear, or responses, such as stigma, mockery, embarrassment, and abandonment:

*‘When you mention those people who come for screening, what comes to our mind when you mention is fear. We start wondering, are these people going to find that we have this disease.’* **Focus group with women, Lomami**

*‘We believe that there are many people who have sleeping sickness, but most run away and refuse to be tested. Some refuse because they will be laughed at.’* **Focus group with men, Kasai Central**


*Question: Is there any reason for not disclosing the identity of those have a positive diagnosis?*


*Answer: Because it is a shameful situation. There is a risk of stigmatization. People fear because some families would abandon or neglect the sick person.* **Focus group with women, Lomami**

Cultural and social norms: Being unable to fulfil certain social expectations, such as bearing children due to HAT, is a reason for concern for many people living in endemic areas: 

*‘I could no longer menstruate, which meant I could no longer have children. I was very disturbed by that’* **Women former patient, Kwilu**

Other social expectations include a person’s ability to be rational or provide for their family. Social and cultural norms also seem to determine the timeliness of access to health care in general, including for HAT, as mentioned earlier concerning gender norms. All these cultural and social norms determine whether people have positive or negative perceptions, attitudes, behaviors, and experiences of HAT. 

HAT endemicity context: most research participants believed that the general context of HAT in their communities has improved compared to the past:

*‘Sleeping sickness exists here, but not like it used to be, the number of cases has dropped because there is a laboratory that was installed here. This has contributed significantly to the reduction of cases.’* **Focus group with women, Kwilu**

According to community members and health providers, it is becoming difficult to get people to participate in screening activities as the number of cases is very low. There are, however, community members and health professionals who do not think that sleeping sickness has decreased. These people attribute the drop in the number of cases to the decrease in the number of people who agree to participate in screening activities, as well as to difficulties in accessing remote or hard to reach places. 

*‘Many cases are still present in the areas that are inaccessible to our mobile teams, and this is an important element to consider. Recently, even in the health zone of Dibanga, 3 or 4 cases were detected in the middle of the city.’* **Key informant, Ministry of Health**

Economic situation and livelihood: Most participants believed that poverty limited their access to quality health services. Local communities perceive that good health care is mostly provided in big hospitals, which are usually far away and expensive. Community members and health professionals raised concerns about the limited success of many activities, such as community education or vector control, because these activities rely on community volunteers. For them, most local people, such as community health workers, cannot afford to volunteer for a long period of time because they are poor. The opportunity cost resulting from participating in screening or accepting treatment is a major factor impeding the success of HAT control activities: 


*‘Many people didn’t come to the mass screening; they were afraid, they were thinking that if they find that they have the sleeping sickness, they will get treated, and they will be asked to stay home during the agriculture season, this would stop their agricultural activities.’*
**Focus group with women, Kasaï**


Policies and strategies related to HAT: Policies and strategies related to community engagement in HAT control activities influence what is happening at community and individual levels. Health professionals at all levels (local, provincial, and national) felt that resources allocated for HAT community education activities were very limited, which also negatively affects the level of adoption of positive behaviors toward HAT:

*‘Another challenge is that people are illiterate. We are having a hard time to make them understand some issues. This requires more time and resources, but currently we have only 2 days planned for our activities per village. Therefore, we have limited time to get a population that is illiterate to understand what we are talking about’* **Key informant, Kwilu**

Both health professionals and alternative health professionals agreed that strategies including collaboration with other health providers such as traditional healers, diviners, and church leaders could reduce the delays in health seeking:

*‘We want that church leaders, traditional healers, and other private health centers don’t keep patients because of ignorance, they should collaborate with health facilities’* **Key informant, health professional, Kasaï Central**

Some senior health professionals and policymakers thought that the high reliance of the government on external funding for its program did not allow for the national HAT control program to set its own priorities. As a result, there are activities that the national control program is not able to carry out even if they think they are important.

## 4. Discussion

Although the study was developed specifically in the most endemic areas of HAT in the DRC, the steep reduction of cases changed the perception of the younger generation and the overall understanding of the risk. Communities’ limited knowledge of the early symptoms is a serious challenge to the timely control of HAT, because if the vector is present, an infected person is a risk for the whole community. Awareness raising is, therefore, needed among community members and frontline workers in terms of identifying early symptoms and referring suspect cases for early detection and treatment. The limited knowledge about the new treatment and the RDT suggests that there have not been enough community education activities on HAT. A study about how local communities sensed the symptoms of HAT in South Sudan found that local communities knew about sleeping sickness, including its symptoms [15]. As in the current study, communities in South Sudan also focused on second stage symptoms. Another study in DRC likewise found that people who had lived in endemic areas had more knowledge about sleeping sickness than those who did not [28]. As result of this study, the PNLTHA, DNDi, and other key stakeholders have used the findings to develop a new communication strategy, which included sensitization materials about HAT. These materials addressed the gaps that were identified in relation to local communities’ knowledge, attitudes, and beliefs about HAT. They were translated into local languages (Tshiluba, Kikongo, and Lingala) before being used for sensitization activities in areas with the highest prevalence of HAT in DRC. 

Generational differences in knowledge about HAT may mean that young people and children do not appreciate the seriousness of the disease. As a result, young people may question the importance of participating in HAT control activities. Age was found to be among factors that contributed to a better knowledge of the sleeping sickness among inhabitants of Kinshasa in DRC [28]. Existing knowledge about HAT among people living in endemic areas should not be taken for granted; if care is not taken to transmit knowledge to the new generation, it could be lost.

Beliefs that HAT could be caused by witchcraft or that the effectiveness of the treatment depends on the source of the disease (natural or supernatural) is a cause of concern for elimination efforts. In a context where witchcraft is rooted in communities’ beliefs, health education alone is not enough. Collaboration with alternative health service providers is a requirement for ensuring that HAT patients are referred early to health centers. However, despite health education, beliefs linking witchcraft with HAT persist in DRC [4,8,9,13]. Similarly in Zambia, communities were found to attribute second stage symptoms of HAT to witchcraft [14]. 

A socio-ecological framework has enabled the disentanglement of the complex factors that contribute to HAT behavior. In this study, these factors were presented at individual, community, and societal levels. 

At the individual level, the perception of risk and vulnerability to HAT, level of knowledge, personal beliefs, health status, personal experience of HAT, age, and gender should not be considered separately. A person’s decision to participate in HAT-related activities usually results from interactions between all these factors. It also appears that some factors weigh more in the decision-making process than others. Identifying which factor weighed the most was not within the scope of this study, and other studies exploring this topic would be of benefit. Personal experience of HAT is an individual factor that appears to contribute to improving community engagement. People who have recovered from HAT can be an asset. As this study has shown, they are consulted by other community members who need to know about HAT. In a context where cases are becoming rare, the perception of HAT as a threat is decreasing. Thus, testimony from former patients can be very convincing for other community members with whom they have many things in common. 

Because individual factors such as age, sex, and gender also have a bearing on participation in HAT control activities, a gender-sensitive approach to HAT community engagement activities is required to address the needs of different groups. In a study using a gender analysis lens, women were found to be less likely to be diagnosed through passive screening compared to men [14]. This contrasts with the results of this current study, which shows that men were less likely to be diagnosed as they were less likely to seek screening or medical care. Gender-based livelihood and cultural activities have been reported to put men more at risk of tsetse bites compared to women [12], but overall evidence on how gender and sex affect HAT behavior is still limited. There is a need for more studies on a gender-sensitive approach to HAT community engagement.

At the interpersonal or community level, the perception of health provision especially influences health-seeking behaviors. While the lack of trust in the motivation of the HAT mobile teams remains one of the barriers to screening, the good quality of health service is appreciated by community members. Lack of trust in mobile teams’ motivation has also been reported by Falisse et.al (2020) [5]. The distance to HAT treatment centers, the poor quality of local health centers, and local health providers lack of expertise about HAT remains a major concern for local communities. Perceptions of the good quality of health services has been reported to motivate local community members to participate in HAT screening, but the indirect costs and the distance to treatment centers are a barrier to agreeing to participate in screening and undergo treatment [11]. However, another study found that finances were the main barrier to accessing health care and that free health services such as free mosquito nets led to suspicions among community members who became reluctant to accept the service [29]. The indirect costs of screening and treatment have also been found to be important factors for participation in HAT control activities [8]. 

In the DRC, where formal social security systems are almost non-existent, the extended family plays an important role in providing the financial support needed to access health care. Families’ knowledge about HAT, beliefs, and financial resources can therefore delay or facilitate access to health services. The family has been shown to play a major role in supporting HAT patients as they bear the cost of health care, and that their lack of knowledge about HAT contributes to delays in accessing health care [30,31]. While a patient is undergoing treatment, the family is responsible for caring for and ensuring that the patient is following medical advice [3]. 

Discrimination of people affected by HAT is not always visible. This could be the reason why stigma surrounding HAT has not been sufficiently considered in management plans. Stigmatization occurs not only within the community but also within the family. Community education on the acceptance of HAT patients is needed, as well as counselling for patients and their families, as is the case with other stigmatizing diseases like HIV/AIDS. People have been found to avoid being screened for HAT as they fear being stigmatized [3]. Stigma associated with HAT was also found in other countries such as Zambia [14] and South Sudan [15]. While it has been suggested that stigma related to HAT does not lead to exclusion [31], this study shows the opposite. Unlike other NTDs, such as leprosy or tuberculosis, stigma about HAT has not been sufficiently studied [23].

The limited ownership of HAT control activities by local communities, as shown by local leaders and other community volunteers increasingly demanding incentives before taking part in HAT activities, has been raised previously [5] and is a concern for community engagement. This situation may have resulted from community members’ perception that their role is limited to vector control, especially clearing vegetation, while health professionals expect them to get involved in screening and treatment. This difference in expectations suggests there is a need for the participation of local people at the planning stage to encourage greater ownership of community-based control activities. 

At the societal level, there are many factors that interact to influence the participation of local communities positively or negatively in HAT control activities. The social representation of HAT is negative, which triggers negative emotions such as fear and shame, which are barriers to screening and treatment. Even though local communities know that HAT can be treated, they also need to see it as a normal disease like malaria or typhoid. In a study [11] in South Sudan, people were found to have a negative perception of HAT, which led to fear and shame; both HAT control activities and the disease were perceived to interfere with the ability of some people to fulfil social expectations (having children, caring for their families or participating fully in community life). All these social expectations need to be clearly discussed with communities. Although the impact of cultural factors, including gender norms, on the differential exposure of women and men to tsetse bites has been investigated [12], there is limited information about how HAT affects social expectations. 

The priority given to livelihood activities over participation in HAT control activities suggests that poverty is high and perceived to be more harmful in the short term than HAT. Economic factors have been found to be a barrier to people participating in screening and accepting treatment [3]. While poverty is difficult to address, HAT control activities can be planned together with community members to ensure that they do not interfere unnecessarily with their livelihood activities. Community education about the fact that there is no need to prohibit heavy labor can reduce concerns related to livelihood activities.

The number of HAT infections in the target provinces of DRC has decreased to the point that control activities, such as active screening, will probably be reduced. Training and equipping community health providers and workers to organize community-led HAT case detection, and referrals could contribute to early detection of HAT at a low cost. In a review of program that used a community-led surveillance approach to tuberculosis [32], it was concluded that in a situation of low prevalence, mass screening would be expensive, and that a community-led approach to surveillance would be more effective and sustainable for early detection, treatment compliance, and fighting stigma. Community-led control has also been found to be effective in monitoring Chagas disease [33]. The same approach could be used for HAT; however, there is a need for more research on how community-led surveillance could be performed, especially in a context of low endemicity. 

Finally, HAT policies and strategies shape what is happening at the community level. The limited resources allocated to community education can hold community members back from benefiting from the advancements made in HAT control (new treatments and RDT). As suggested in this study, people are still avoiding screening and/or treatment due to a lack of adequate information. It is, therefore, important to allocate sufficient resources (financial, human, and time) to community education when developing HAT control strategies. In the context of other diseases, such as malaria [34] or HIV/AIDS [35], policies that allocate sufficient resources and empower local communities seem to yield positive results in terms of community engagement. There seems, however, to be a gap in such community engagement in relation to HAT.

There were some limitations to this study. Data analysis was not disaggregated per province, which makes it difficult to capture the peculiarities that are specific to each province. Additionally, the findings can only apply to regions of DRC where HAT control activities, such as mass screening or vector control activities, are still visible. 

The strength of this study lies in its methodological approach. The research team spent on average three days in each community, giving time for community members to understand the purpose of the study and to get used to the research team. Despite the challenges related to working in DRC (vastness of the country, rainy season, and poor road infrastructure), this study covered six provinces (Kasai, Kasai Central, Kasai Oriental, Lomami, Maindombe, and Kwilu) with diverse ecosystems, cultures, and linguistic groups. The sampling plan allowed the research team to gather information from a wide range of participants.

## 5. Conclusions

With HAT cases becoming increasingly rare, and with the related decrease in active screening, it is desirable for local communities to take a more active role in HAT control activities. As shown in this study, local communities are knowledgeable about HAT. While many community members have understood the importance of screening, treatment, or vector control, their participation in these activities remains insufficient to facilitate the elimination of the disease. To enhance community engagement, it is important to understand the factors that influence communities’ behavior. By using a socio-ecological model, this study found that these factors can be identified at three levels: individual (age, gender, health status, experience of the disease, knowledge, belief, and perception of the risk), community or interpersonal (health service provision, family, stigma, local leadership, education, and sensitization activities), and societal levels (social representation of HAT, culture and social norms, HAT endemicity, economic situation and livelihood, and policies and strategies related to community engagement).

This study has many implications. Firstly, health professionals need to see HAT patients not only as individuals, but also as people living in a specific context that affects their personal circumstances. Secondly, policymakers need to recognize that the success of HAT control strategies depends on effective community engagement. Thirdly, factors need to be addressed at the three levels of individuals, communities, and societies. Given that a reduced number of cases of HAT will most probably result in reduced levels of funding, there is an urgent need to find more sustainable and efficient ways to continue HAT surveillance. If well-equipped and well-prepared communities living in endemic areas could increase early identification and referrals of suspected cases, this would play a key part in DRC interrupting the transmission of HAT.

## Figures and Tables

**Figure 1 diseases-10-00069-f001:**
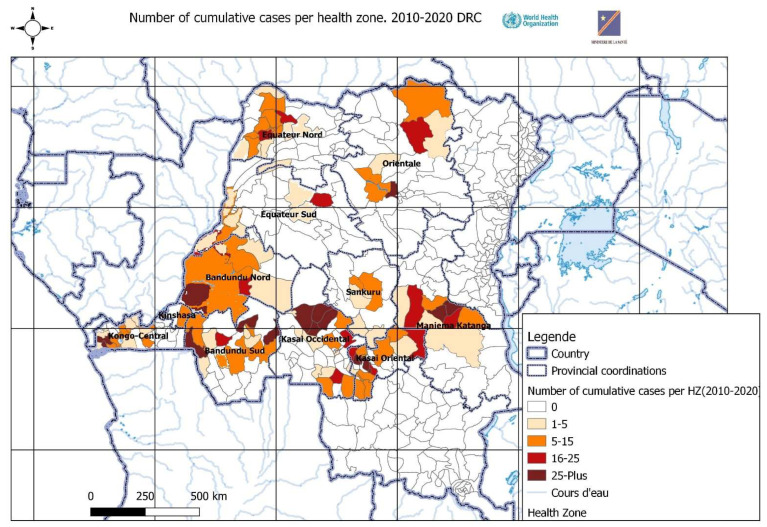
Map of HAT prevalence in the Democratic Republic of Congo. (Source = PNLTHA DRC).

**Figure 2 diseases-10-00069-f002:**
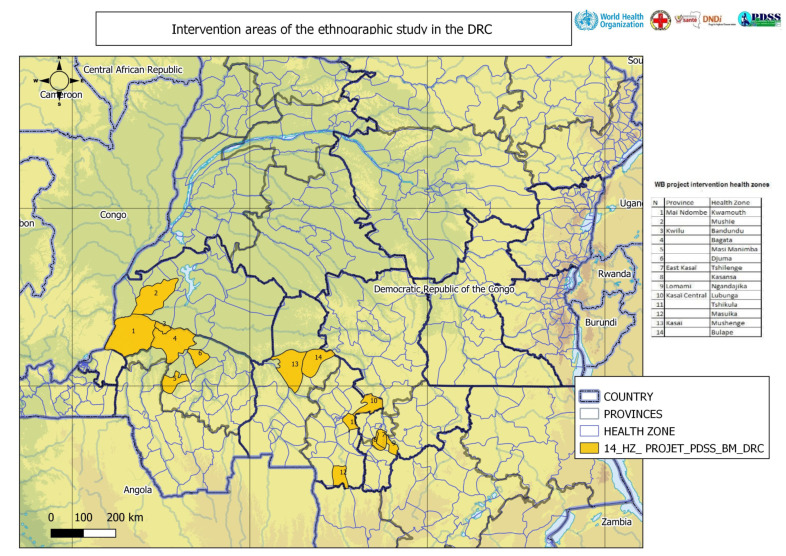
Map of the health zones studied. (Source = PNLTHA DRC).

**Figure 3 diseases-10-00069-f003:**
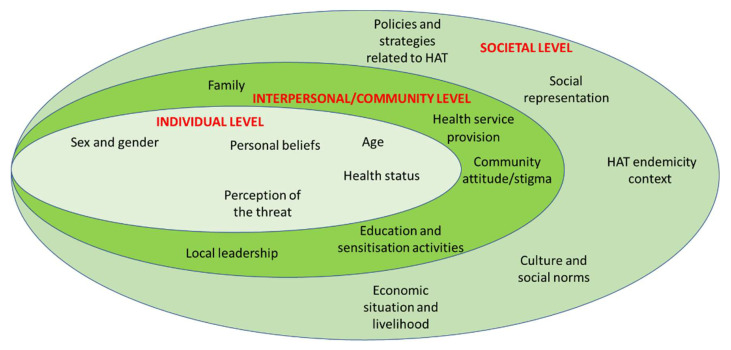
Factors influencing HAT related behaviors: Socio-ecological model. Adapted from Grandner, M. A. (2019) [27].

**Table 1 diseases-10-00069-t001:** Data collection activities and participants.

Province	Health Zones	Data Collection Activities				
		Focus Group Discussions	Key Informant Interview	Life Story	Participatory Workshop	Observation ^1^
Kwilu	Bagata	3	5	3	1	Continuous
	Masimanimba	2	4	2	1	
	Bandundu	3	5	3	1	
	Djuma	3	6	1	1	
Kasai	Bulape	3	2	3	1	
	Mushenge	1	5	3	1	
Maindombe	Kwamouth	3	5	4	1	
	Mushie	3	7	2	1	
Lomami	Ngandajika	3	5	2	1	
Kasai Oriental	Tshilenge	3	5	3	1	
	Kasansa	2	4	1	1	
	Mbujimayi		5			
Kasai Central	Tshikula	3	4	2	1	
	Masuika	3	2	1		
	Lubunga	4	4	4	1	
	Kananga		2			
Kinshasa			3			
Total		39	73	34	13	

^1^ Observation sites included churches, traditional healers, village surroundings, water points, etc.

## Data Availability

The data underlying the results of this study are available upon request because they contain potentially sensitive information. Interested researchers may contact the Drugs for Neglected Diseases *initiative* (DNDi), commissioner of this study, for data access requests via email at CTdata@dndi.org. Researchers may also see DNDi commitment for transparency and request data by completing the form available at https://www.dndi.org/category/clinical-trials/ (last accessed 19 September 2022). In this, they confirm that they will share data and results with DNDi and will publish any results open access.

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
