# Peer review of "Communities’ Perception, Knowledge, and Practices Related to Human African Trypanosomiasis in the Democratic Republic of Congo"

_diseases, 2022, doi:10.3390/diseases10040069_

Round 1
Reviewer 1 Report
Dear Authors,
The paper you submitted proposes using a socio-ecological model to identify the factors influencing the DRC communities' behaviour towards Human African trypanosomiasis. The cases are becoming rare but there is still considerable work to do to reach the elimination of HAT.
I found your work well designed, carried out and also well presented in this paper, and it will be a piece of great interest to the scientific community.
Author Response
Dear Reviewer.
We are grateful for your favourable opinion about our article. We have introduced several modifications to the english language to try to improve understandability. We added one table and two maps to improve the information about the methodology of the study.
Warm regards
Olaf Valverde Mordt. Corresponding author
Reviewer 2 Report
Thank you for your work investigating perception, knowledge and practices in the Democratic Republic of Congo (DRC) with regards to Human African Trypanosomiasis (HAT). The manuscript is well written and methods are valid to investigate the science being conducted. I appreciate the direct translated quotations from respondents and this approach likely strengthened the utility of the investigation. This allows the readers to understand more deeply how HAT is perceived and disparities that exist.
One recommendation I have is describing in the Material and Methods section more details pertaining to the locations (villages, cities, provinces) where the study was conducted as it pertains the data collection type (PW, FGD, IDIs, KIIs, observations). I recommend a GIS style figure to accompany this information. I also recommend the figure to have show known endemic regions of the country and possibly known cases or regions which high prevalence. HAT is predicted to possibly occur throughout DRC (vector found throughout) but there are regions with much higher prevalence of infection within the country. Providing this information would be impactful.
I would also mention this in the Discussion section. If the study was conducted in the regions with higher prevalence, then targeted interventions are recommended based off the findings.
Author Response
Dear Reviewer.
We are grateful for your favourable opinion about our article and your useful comments. We have introduced several modifications to the English language to try to improve understandability. We added one table and two maps to improve the information about the methodology and location of the study. We are still preparing a better-quality map as the ones accompanying this answer are low resolution, but we wished to answer rapidly while preparing the final version of the article, so that you could check if the type of maps chosen are satisfactory. We also commented in the discussion about our focus in the geographical areas with higher endemicity and the practical outcome of our study in the production of new communication materials for the communities, together with the Sleeping Sickness National Control Programme of the DRC.
Warm regards
Olaf Valverde Mordt. Corresponding author